# Clinical Response and Quality of Life in Patients with Severe Atopic Dermatitis Treated with Dupilumab: A Single-Center Real-Life Experience

**DOI:** 10.3390/jcm9030791

**Published:** 2020-03-13

**Authors:** Silvia Ferrucci, Giovanni Casazza, Luisa Angileri, Simona Tavecchio, Francesca Germiniasi, Emilio Berti, Angelo Valerio Marzano, Giovanni Genovese

**Affiliations:** 1UOC Dermatologia, Fondazione IRCCS Ca’ Granda Ospedale Maggiore Policlinico, 20122 Milan, Italy; silvia.ferrucci@policlinico.mi.it (S.F.); luisa.angileri@unimi.it (L.A.); simona.tavecchio@gmail.com (S.T.); francesca.germiniasi@unimi.it (F.G.); emilio.berti@unimi.it (E.B.); angelo.marzano@unimi.it (A.V.M.); 2Dipartimento di Scienze Biomediche e Cliniche “L. Sacco”, Università degli Studi di Milano, 20157 Milan, Italy; giovanni.casazza@unimi.it; 3Dipartimento di Fisiopatologia Medico-Chirurgica e dei Trapianti, Università degli Studi di Milano, 20122 Milan, Italy

**Keywords:** atopic dermatitis, dupilumab, quality of life, disease severity

## Abstract

Dupilumab is an anti-interleukin-4 receptor monoclonal antibody that was recently approved for the treatment of atopic dermatitis (AD). In this single-center retrospective study, clinical baseline data of 117 severe AD patients treated with dupilumab were collected. At baseline and at weeks 4 and 16, disease severity was assessed through the Eczema Area and Severity Index (EASI) and quality of life through the Dermatology Life Quality Index (DLQI) questionnaire, Patient-Oriented Eczema Measure (POEM), Hospital Anxiety and Depression Scale (HADS), Peak Pruritus Numerical Rating Scale (NRS-itch), and VAS-sleep. Response to dupilumab was defined as an improvement of ≥75% in EASI from baseline (EASI75). At multivariate analysis, AD onset before 18 years [OR, 2.9; 95% CI, 1.2–7.2; *p* = 0.0207] and absence of hypereosinophilia [OR, 2.24; 95% CI, 1.03–4.86; *p* = 0.0412] were identified as significant predictive parameters for response to dupilumab in terms of EASI75 at week 4 but not at week 16. Significant reductions in EASI, DLQI, POEM, HADS, NRS-itch, and VAS-sleep were found between week 4 versus baseline (*p* < 0.0001 for all) and week 16 versus baseline (*p* < 0.0001 for all). Early AD onset and absence of hypereosinophilia may be suggested as predictive markers of early response to dupilumab. We confirmed the efficacy and safety of this agent along with the improvement of life quality in severe AD patients.

## 1. Introduction

Atopic dermatitis (AD) is a debilitating, chronic-relapsing inflammatory dermatosis characterized by skin xerosis, pruritic lesions, and detrimental effects on sleep, mood, productivity, and quality of life [1]. Its complex pathophysiology involves both the disruption of skin barrier function and a T helper 2 (Th2)-polarized immune response [2]. Management of mild forms of AD relies on the use of emollients, topical corticosteroids/calcineurin inhibitors, and phototherapy, while systemic immunomodulant/immunosuppressive agents such as corticosteroids and cyclosporine A (CsA) are used for severe refractory cases. Looking beyond conventional drugs, data on the newer targeted therapies, including monoclonal antibodies and oral small molecules, are very promising [3,4,5]. In this regard, dupilumab (Dupixent^®^), a fully human monoclonal antibody inhibiting the interleukin (IL)-4/IL-13 signaling through the blockade of the IL-4 receptor α subunit, has been recently approved by both the US Food and Drug Administration (FDA) and the European Medicines Agency (EMA) for the treatment of patients with moderate-to-severe inadequately controlled AD [6]. However, although dupilumab has been demonstrated to be an effective and safe therapeutic option by the two identical phase-3 SOLO1 and SOLO2 trials [7] and the LIBERTY AD CHRONOS [8] and LIBERTY AD CAFÉ [9] phase-3 trials, literature data resulting from a real-life daily practice setting are limited [10,11,12,13,14,15,16,17,18].

Therefore, with the aim of identifying possible predictors of response to dupilumab and assessing the clinical response to this drug in terms of quality of life and disease severity improvement, we retrospectively assessed cases of severe AD treated with dupilumab for a period of at least 16 weeks at our Department.

## 2. Materials and Methods

### 2.1. Patients

We performed a retrospective chart review of 117 patients with severe AD treated with dupilumab for a period of at least 16 weeks from June 2018 to November 2019 at the Dermatology Department of the University of Milan, Italy. According to the Italian Drug Agency (AIFA) recommendations for dupilumab prescription, inclusion criteria were: (i) age ≥18 years; (ii) severe disease defined by a baseline Eczema Area and Severity Index (EASI) ≥24; (iii) inadequate response/intolerance to CsA or medical inadvisability of CsA treatment. Patients with any documented psychiatric comorbidity were excluded from the study.

All patients were treated with self-administered subcutaneous dupilumab 300 mg every other week following a loading dose of dupilumab 600 mg subcutaneously administered by a clinician. Traditional immunosuppressive agents (e.g., CsA, azathioprine, methotrexate) were discontinued at least 4 weeks before dupilumab initiation in all patients, while systemic corticosteroids were maintained in a minority of patients, with progressive tapering and subsequent withdrawal within 2 weeks. Concomitant topical corticosteroids or calcineurin inhibitors were allowed. Patients were evaluated three times: at baseline, 4, and 16 weeks after dupilumab initiation.

All patients agreed with the treatment regimen and signed a written consent form to extract relevant data from their charts. In view of the retrospective nature of the study, only a notification to the Ethical Committee of the Fondazione IRCCS Ca’ Granda Ospedale Maggiore Policlinico, Milan Italy) was requested.

### 2.2. Demographic, Clinical, and Laboratory Features

At baseline, the following data were collected: sex, age at AD onset, age at dupilumab initiation, extrinsic/intrinsic status, baseline total serum immunoglobulin (Ig)E, baseline serum eosinophil count, previous treatment with CsA (intolerance/ineffectiveness/contraindication), and concurrent use of systemic corticosteroids at baseline. Adult-onset AD was defined by the cut-off value of 18 years, while hypereosinophilia was defined in presence of blood eosinophil count > 500 × 10^3^/L. Patients were categorized in the intrinsic AD group in cases of: (i) absence of other atopic diseases such as allergic asthma and rhinoconjunctivitis, (ii) negative prick and/or intracutaneous skin tests for common inhalant and food allergens, and (iii) total serum IgE levels ≤ 200 kU/L. Total serum IgE were re-assessed 4 weeks after dupilumab initiation and the ratio between week 4 and baseline total serum IgE was calculated.

### 2.3. Physician- and Patient-Reported Outcomes

Disease severity was determined at baseline and at weeks 4 and 16 after dupilumab, starting by means of the physician-reported outcome EASI [19]. Response to dupilumab was defined as an improvement of ≥75% in EASI from baseline (EASI75). Additionally, patient-reported outcomes including the Italian version of Dermatology Life Quality Index (DLQI) questionnaire [20], Patient-Oriented Eczema Measure (POEM) [21], Hospital Anxiety and Depression Scale (HADS) [22], Peak Pruritus Numerical Rating Scale (NRS-itch) during the past 7 days [23], and VAS-sleep [24], which are measurement instruments widely accepted to evaluate quality of life in AD, were collected at baseline and at weeks 4 and 16 after dupilumab initiation.

### 2.4. Statistical Analysis

Categorical variables were reported as frequencies and percentages while continuous variables were reported as medians and interquartile range (IQR). The non-parametric Wilcoxon signed-rank test was used to compare paired values of total serum IgE, physician-reported, and patient-reported outcomes between week 4 and baseline and between week 16 and baseline.

Logistic regression models were used to assess the association between response to dupilumab (EASI reduction >75%) and the following potential predictive factors: sex; age at onset (dichotomised as < or >18 years); age at dupilumab initiation (dichotomised as < or > observed median value); previous CsA treatment; status (intrinsic versus extrinsic); hypereosinophilia (>500 eosinophils/μL); baseline total IgE (dichotomised as < or > observed median value); and concurrent therapy with systemic corticosteroids. Univariate and multivariate logistic regression analyses were performed. At the first step, univariate models were fitted, considering all the variables reported above, in order to identify potentially relevant predictors to be considered in multivariate analyses. A less-restrictive *p* value < 0.10 was used to identify candidate predictors to be included in the multivariate analysis. Finally, a multivariate model was fitted considering only the variables identified at the univariate step. Two separate logistic regression analyses were performed considering EASI reduction at week 4 and at week 16.

*P* values less than 0.05, two sided, were considered statistically significant. The statistical software SAS (release 9.4, SAS Institute, Inc., Cary, NC, USA) was used to perform all the statistical analyses.

## 3. Results

### 3.1. Patients’ Demographic Data and Predictors of Response

A total of 117 patients, 52 females and 65 males, met the inclusion criteria and were eligible for the study. The patients’ characteristics are summarized in Table 1. They had a mean age at onset of 11 years (range: 0–76) and a median age at dupilumab initiation of 39 (IQR: 26-50). Adult-onset AD was observed in 31 (26.5%) patients, while an intrinsic type of AD was observed in 19 (16.2%) patients. Prior to dupilumab, CsA had been administered in 103 patients, of whom 49 (41.9%) developed adverse events and 54 (46.1%) showed an inadequate response. Conversely, 14 patients had never received CsA as it was contraindicated.

Median baseline eosinophil count was 350/μL (IQR: 120–690/μL) and 50 (42.7%) patients were classified as having hypereosinophilia. Median baseline total IgE were 1809 kU/L (IQR: 338–4210 kU/L; mean: 4210.2 kU/L) and median week 4 total IgE were 1938 (IQR:242–4680 kU/L; mean 3602.3 kU/L) [see Table 1]. Median ratio between baseline and week 4 total IgE was 1.27 (IQR:1.08–1.65; mean: 1.89). Twenty-three (19.7%) patients were maintained on systemic corticosteroids during dupilumab treatment, with progressive tapering and steroid discontinuation within 3 weeks in all cases. The statistical analysis showed no association between w4IgE/bIgE ratio and the EASI variation between baseline and week 4 (*p* = 0.21).

As shown in Table 2, sex, intrinsic versus extrinsic status, baseline serum total IgE (dichotomised using the median value, that is < or >= 1809 kU/L), early dupilumab initiation, and intolerance/contraindication versus ineffectiveness of CsA were not important predictors of response to dupilumab in terms of EASI75 either at week 4 and at week 16. Interestingly, at multivariate analysis early AD onset [OR, 2.9; 95% CI, 1.2–7.2; *p* = 0.0207] and absence of hypereosinophilia [OR, 2.24; 95% CI 1.03–4.86; *p* = 0.0412)] were identified as significant predictive parameters for response to dupilumab in terms of EASI75 at week 4. At week 16, no predictive parameters were significantly associated with response to dupilumab. Considering the multivariate model including the two factors with *p* < 0.1 at univariate analysis at week 4, we found that early AD onset confirmed the statistical significance at multivariate analysis.

### 3.2. Disease Severity

Median baseline EASI was 30 (IQR: 24–37) and significantly dropped to 8 (IQR: 5–12; *p* < 0.001) at week 4. A further reduction of median EASI was observed at week 16, when its value was 5 (IQR: 2–8). This reduction was statistically significant with respect to both week 4 (*p* < 0.001) and baseline (*p* < 0.001) [see Table 1]. Fifty-four out of 117 (46.2%) patients achieved EASI75 at week 4, while at week 16, patients who achieved EASI75 raised to 85/117 (72.7%).

### 3.3. Quality of Life

As shown in Table 3, the median DLQI score, which was 16 at baseline, dropped to 5 at week 4 and to 3 at week 16. The median POEM score, which was 23 at baseline, dropped to 9 at week 4 and to 6 at week 16. We found a statistically significant reduction in DLQI scores between week 4 *versus* baseline (*p* < 0.0001) and week 16 versus baseline (*p* < 0.0001). Likewise, we found a statistically significant reduction in POEM scores of week 4 versus baseline (*p* < 0.0001) and week 16 versus baseline (*p* < 0.0001). Median HADS-anxiety at baseline was 8 (IQR:5–11) and dropped to 4 (IQR: 2–6) at week 4 and to 3 (IQR: 1–5) at week 16. Similarly, HADS-depression dropped from 7 (IQR: 4–10) at baseline to 4 (IQR: 1–7) at week 4 and to 3 (IQR: 0–6) at week 16. Median NRS-itch, which was 9 (IQR: 8–10) at baseline, dropped to 4 (IQR: 2–5) at week 4 and to 3 (IQR: 1–4) at week 16, while median VAS-sleep was 8 (IQR: 5–10) at baseline and dropped to 1 (IQR: 0–3) at week 4 and to 0 (IQR: 0–1) at week 16. Statistically significant differences were observed either for HADS-anxiety, HADS-depression, NRS-itch, and VAS-sleep when comparing values of week 4 versus baseline (*p* < 0.0001) and week 16 versus baseline (*p* < 0.0001).

### 3.4. Safety

The majority of adverse events were mild in severity and included blepharoconjunctivitis (*n* = 14; 11.9%), facial redness (*n* = 6; 5.1%), and paradoxical psoriasis (*n* = 1; 0.8%). The severity of the adverse event led to drug discontinuation only in a patient with blepharoconjunctivitis and in the patient with paradoxical psoriasis at week 24 and week 32, respectively.

## 4. Discussion

This retrospective single-center study on a cohort of 117 patients with severe AD reflects the Italian real-life experience in the management of the disease with dupilumab and confirms the efficacy of this agent for refractory cases. Indeed, considering that our study population is similar to that of the LIBERTY AD CAFE study, being represented by patients with severe AD refractory to CsA, approximately 70% of our patients, in line with most studies present in the literature [8,9,10], achieved EASI75 at week 16 after starting dupilumab.

A statistically significant reduction in EASI score was achieved from baseline at week 4 and week 16 after starting dupilumab. As of now, some of the patients included in the study achieved week 52, maintaining a good response and low EASI score (unpublished data). The clinical improvement resulting from the physician-reported outcome EASI was accompanied by a considerable amelioration of quality of life, witnessed by a significant reduction of patient-reported outcomes widely used in AD such as DLQI, POEM, NRS-itch, and VAS-sleep from baseline both at week 4 and week 16 [25,26,27]. In addition, we investigated anxiety and depression using HADS and we substantiated, in accordance with a previous study by Cork et al. [28], a significant reduction in anxiety and depression scores after dupilumab administration.

The most remarkable finding in our study is that patients with an early AD onset seemed to respond better to dupilumab at week 4. The confirmation of this result in the multivariate analysis, where we decided to include variables with *p* value < 0.1 in the univariate analysis (instead of the usual value of *p* < 0.05), emphasizes the importance of early AD onset as a predictor of response to dupilumab. Even if at week 16 we failed to confirm this association, early onset showed a certain association (OR = 1.7) with response in terms of EASI75, albeit the result was not statistically significant. This may be due to the relatively small sample size, with consequent low statistical power.

Furthermore, the result indicating absence of hypereosinophilia as a predictive biomarker of response (OR = 2.24; *p* = 0.0412) in the multivariate analysis at week 4 suggests a possible role of eosinophils in response to dupilumab. Indeed, inhibition of group 2 innate lymphoid (ILC2) cells and, consequently, eosinophils may be one of the mechanisms of action of dupilumab [29]. However, further experimental and clinical studies are needed to confirm this hypothesis.

In our cohort, extrinsic status and baseline total IgE was not correlated with response to dupilumab both at week 4 and at week 16. Moreover, as expected [30], although a slight decrease of total IgE was observed in the first 4 weeks of treatment, the ratio between week 4 and baseline total IgE did not correlate with dupilumab response in terms of EASI75.

Finally, we also showed for the first time that patients who experienced ineffectiveness of a previous course of CsA had similar response to dupilumab as compared to patients with contraindication or intolerance to this drug, suggesting that response to CsA does not influence the effectiveness of dupilumab.

Dupilumab was well tolerated in most patients, with only 11.9% of patients suffering from blepharoconjunctivitis, a finding which is in contrast to studies reporting eye symptoms in up to 62% of cases [18]. Although the underlying mechanism of blepharoconjunctivitis in AD patients is still not completely elucidated, some authors have suggested that the blockade of IL-4 and IL-13 may increase the activity of specific ligands, such as OX40 ligand, involved in atopic keratoconjunctivitis [31]. It can be assumed that the relatively low incidence of this side effect might be due to the systematic use of lipid emulsion eye drops combined with hyaluronic acid eye drops in all patients from the first day of dupilumab. All patients with blepharoconjunctivitis were seen by an ophthalmologist when symptoms occurred. Blepharoconjuctivitis was the most common adverse event associated with dupilumab and according to the literature [11,32], it was successfully managed with chloramphenicol/bethametasone 0.2 + 0.5% eye drops. In resistant cases, we introduced tacrolimus 0.03% eye ointment. Blepharoconjuctivitis led to drug withdrawal only in cases resistant to topical treatment. Injection-site reactions were not reported by our patients. Limitations of the study are its retrospective nature, the small sample size, and the short follow-up. We are aware of the fact that these data may differ from other AD referral centers.

To sum up, our real-life study confirmed that dupilumab is an effective treatment in most patients with severe AD and identified early AD onset and absence of hypereosinophilia as predictive markers of early response to this agent.

## Figures and Tables

**Table 1 jcm-09-00791-t001:** Demographic and clinical features of the patients.

	*n* = 117
Males, *n* (%)	65 (55.6)
Age at onset, years, mean (range)	11 (0–76)
Age at dupilumab initiation, median (IQR)	39 (26–50)
Early onset, *n* (%)	86 (73.5)
Hypereosinophilia, *n* (%)	50 (42.7)
Extrinsic AD, *n* (%)	98 (83.8)

**Table 2 jcm-09-00791-t002:** Odds ratios (OR) and 95% confidence intervals (CI) for response to dupilumab in terms of EASI75 according to baseline clinical parameters in patients with severe atopic dermatitis at week 4 and week 16.

		Week 4 §			Week 16		
		OR	95% CI	*p* Value	OR	95% CI	*p* Value
Sex	Females	1 *		1	1 *		0.355
	Males	1	0.5–2.1		0.7	0.3–1.6	
Early/adult-onset (< or >18 years)	Adult-onset	1 *		0.029	1 *		0.239
	Early-onset	2.7	1.1–6.5		1.7	0.7–4.1	
Age at dupilumab initiation	>39	1 *		0.493	1 *		0.585
	<=39	1.3	0.6–2.7		0.8	0.4–1.8	
Previous cyclosporine A treatment	Contraindication or intolerance	1 *		0.728	1 *		0.251
	Ineffectiveness	1.2	0.6–2.4		1.6	0.7–3.7	
Intrinsic versus extrinsic status	Intrinsic	1 *		0.7	1 *		0.652
	Extrinsic	1.2	0.5–3.3		1.3	0.4–3.7	
Hypereosinophilia (>500 eosinophils/μL)	Yes	1 *		0.059	1 *		0.579
	No	2.1	1–4.4		1.3	0.6–2.9	
Baseline total IgE (>1809 KU/L)	No	1 *		0.776	1 *		0.440
	Yes	0.9	0.4–1.9		1.4	0.6–3.1	
Concurrent therapy with systemic corticosteroids	No	1 *		0.347	1 *		0.891
	Yes	1.5	0.7–3.3		1.3	0.4–2.6	

* Reference category. § At multivariate analysis, including “early/adult-onset” and “hypereosinophilia”, the following results were obtained: “early/adult-onset”: OR for early onset: 2.91 (95% CI 1.18–7.18; *p* = 0.0207), “hypereosinophilia”: OR for absence of hypereosinophilia: 2.24 (95% CI 1.03–4.86; *p* = 0.0412).

**Table 3 jcm-09-00791-t003:** Total serum IgE, physician-reported, and patient-reported outcomes at baseline, week 4, and week 16 after dupilumab starting.

	Baseline	Week 4	*p* Value *	Week 16	*p* Value **
Serum total IgE	1809 (338–5434)	1938 (242–4680)	<0.0001	–	–
EASI	30 (24–37)	8 (5–12)	<0.0001	5 (2–8)	<0.0001
POEM	23 (18–27)	9 (5–12)	<0.0001	6 (3–10)	<0.0001
DLQI	16 (12–22)	5 (3–10)	<0.0001	3 (1–7)	<0.0001
HADS-depression	7 (5–10)	4 (2–7)	<0.0001	3 (1–5.5)	<0.0001
HADS-anxiety	8 (5–11)	4 (2–6)	<0.0001	3 (1–5)	<0.0001
NRS-itch	9 (8–10)	4 (2–5)	<0.0001	3 (1–4)	<0.0001
VAS sleep	8 (5–10)	1 (0–3)	<0.0001	0 (0–1)	<0.0001

Data are reported as median (interquartile range). * comparison between week 4 and baseline. ** comparison between week 16 and baseline.

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
