# Peer review of "Clinical Response and Quality of Life in Patients with Severe Atopic Dermatitis Treated with Dupilumab: A Single-Center Real-Life Experience"

_jcm, 2020, doi:10.3390/jcm9030791_

Round 1
Reviewer 1 Report
In this single-center retrospective study, authors collected clinical baseline data of 117 severe AD patients treated with dupilumab, assessed the EASI, DLQI, POEM, HADS, NRS-itch and VAS-sleep indexes scientifically. Authors found significant reductions in EASI, DLQI, POEM, HADS, NRS-itch after dupilumab treatment, suggested that early AD onset and absence of hypereosinophilia may be predictive markers of early response to dupilumab. Authors also confirmed the efficacy and safety of dupiluimab. This is a very logic and scientific work. Here are some comments below:
- Nowadays, more and more papers report that innate lymphoid cell type 2 (ILC2) is very import for the developing of atopic dermatitis. ILC2 secretes Th2 cytokine, such as IL-4, IL-5, IL-9, IL-13. IL-5 could activate eosinophils. In this paper, authors reported dupilumab diminished the hypereosinophilia in early AD onset patients. Do you thinks dupilumab partly work via inhibition of ILC2 cells? Authors could also detect the ILC2s in the blood. Besides, why dupilumab only diminished the hypereosinophilia in early AD onset patients?
- Patients part: “from June 2018 to November 2019 at the Dermatology 61 Department of the University of Milan, Italy”. Just to remind that this was summer and autumn, people usually wear little clothes, getting exposed to environment, which could possiblely affect the result.
- Patients part: “All patients were treated with self-administered subcutaneous dupilumab 300 mg every other week following a loading dose of dupilumab 600 mg subcutaneously administered by a clinician”. Why the dose is different? Why choose subcutaneous injection this antibody, because commonly abs are ip or iv administrated? Besides, where is the subcutaneous place, lesional part or healthy part?
- Patient part: “Patients were evaluated three times: at baseline, 4 and 16 weeks after dupilumab initiation”. According to authors results (table 2 and 3), dupilumab was proved to be effective in reducing the severity of AD. However, there seems to be no big difference between 4 weeks and 16 weeks. Does this mean duplimab works in a fast way by neutralize IL-4? Does author has the data of week 2? I am also very curious about the result of long time using (> 6 month), if there is no significant difference, it may be due to the tolerance.
- Safety part: “The majority of adverse events were mild in severity and included blepharoconjunctivitis (n=14; 11.9%), facial redness (n=6; 5.1%), and paradoxical psoriasis”. This is a high percentage of blepharoconjunctivitis, what do you think the possible reason caused it?
- Discussion part: “In our cohort, extrinsic status and baseline total IgE did not correlated to response to dupilumab both at week 4 and at week 16. This is in contrast with the results of Olesen et al. [10] who described high baseline total IgE as predictors of non-response. How to explain this part?
Author Response
Comment: In this single-center retrospective study, authors collected clinical baseline data of 117 severe AD patients treated with dupilumab, assessed the EASI, DLQI, POEM, HADS, NRS-itch and VAS-sleep indexes scientifically. Authors found significant reductions in EASI, DLQI, POEM, HADS, NRS-itch after dupilumab treatment, suggested that early AD onset and absence of hypereosinophilia may be predictive markers of early response to dupilumab. Authors also confirmed the efficacy and safety of dupiluimab. This is a very logic and scientific work.
Reply: We thank the Reviewer for appreciating our manuscript.
Here are some comments below:
Comment: Nowadays, more and more papers report that innate lymphoid cell type 2 (ILC2) is very import for the developing of atopic dermatitis. ILC2 secretes Th2 cytokine, such as IL-4, IL-5, IL-9, IL-13. IL-5 could activate eosinophils. In this paper, authors reported dupilumab diminished the hypereosinophilia in early AD onset patients. Do you thinks dupilumab partly work via inhibition of ILC2 cells? Authors could also detect the ILC2s in the blood. Besides, why dupilumab only diminished the hypereosinophilia in early AD onset patients?
Reply: We thank the Reviewer for giving us the opportunity to clarify this point. Since eosinophil count was collected only at baseline (and not at week 4 or week 16), we cannot provide conclusions on the effect of dupilumab on eosinophils. We have simply identified early AD onset and absence of hypereosinophilia as predictive markers of early response to dupilumab. The importance of eosinophils as markers of response to dupilumab is one of the most significant take-home messages of this paper. Indeed, patients with high concentration of eosinophils tend to not respond to dupilumab. Inhibition of ILC2 cells may be surely one of the mechanisms of action of dupilumab (as demonstrated by Patel et al. who found that in a subset of patients with severe atopic dermatitis on dupilumab there was reduced frequency of ILC2 compared to healthy controls, suggesting that inhibition of IL-4 and IL-13 signaling through IL-4Ra may lead to decrease frequency of ILC2 [Patel, Gargi et al. Group 2 Innate Lymphoid Cells in Patients with Severe Atopic Dermatitis on Dupilumab Journal of Allergy and Clinical Immunology, Volume 143, Issue 2, AB19]). We have added the following sentence in the discussion section: “Indeed, inhibition of group 2 innate lymphoid (ILC2) cells and, consequently, eosinophils may be one of the mechanisms of action of dupilumab [29].”. Finally, unfortunately we cannot detect ILC2s in the blood owing to the retrospective nature of this study, but it could be matter for future investigations.
Comment: Patients part: “from June 2018 to November 2019 at the Dermatology 61 Department of the University of Milan, Italy”. Just to remind that this was summer and autumn, people usually wear little clothes, getting exposed to environment, which could possiblely affect the result.
Reply: The enrolment period lasted 17 months, from June 2018 to November 2019. Thus, patients were recruited and followed-up in different seasons (summer and autumn but also winter and spring).
Comment: Patients part: “All patients were treated with self-administered subcutaneous dupilumab 300 mg every other week following a loading dose of dupilumab 600 mg subcutaneously administered by a clinician”. Why the dose is different? Why choose subcutaneous injection this antibody, because commonly abs are ip or iv administrated? Besides, where is the subcutaneous place, lesional part or healthy part?
Reply: All patients were treated with the same standard protocol for dupilumab, which consist of 300 mg every other week following a loading dose of dupilumab 600 mg. The loading dose is fixed and standardized for all patients. Subcutaneous injections were performed in the arm and always in an area spared by the disease. Dupilumab, as well as other monoclonal antibodies used in dermatology such as omalizumab and adalimumab, is administered subcutaneously.
Comment: Patient part: “Patients were evaluated three times: at baseline, 4 and 16 weeks after dupilumab initiation”. According to authors results (table 2 and 3), dupilumab was proved to be effective in reducing the severity of AD. However, there seems to be no big difference between 4 weeks and 16 weeks. Does this mean duplimab works in a fast way by neutralize IL-4? Does author has the data of week 2? I am also very curious about the result of long time using (> 6 month), if there is no significant difference, it may be due to the tolerance.
Reply: Certainly, dupilumab rapidly improves skin symptoms of patients with atopic dermatitis. Subsequently, even if slower and less marked, the improvement is steady. At this moment (March 2020), we are following-up some patients (unpublished data) that are at week 52 after dupilumab starting: in these patients, dupilumab allowed to achieve excellent results in terms of EASI reduction. The following sentence has been added in the discussion section: “As of now, some of the patients included in the study achieved week 52, maintaining a good response and low EASI score (unpublished data)”. Unfortunately, patients came back to the inpatient service only after 4 weeks from dupilumab initiation. Therefore, we have no data regarding week 2.
Comment: Safety part: “The majority of adverse events were mild in severity and included blepharoconjunctivitis (n=14; 11.9%), facial redness (n=6; 5.1%), and paradoxical psoriasis”. This is a high percentage of blepharoconjunctivitis, what do you think the possible reason caused it?
Reply: Literature data report a prevalence of blepharoconjunctivitis in patients with atopic dermatitis treated with dupilumab which is higher than that reported in our study . In our cohort of patients, systematic use of lipid emulsion eye drops combined with hyaluronic acid eye drop might have prevented the development of this side effect. Although the underlying mechanism of blepharoconjunctivitis in these cases are still unclear, dupilumab use for asthma or nasal polyposis was not associated to have higher rates of conjunctivitis. This suggests a unique relationship between dupilumab use for atopic dermatitis and ocular complications rather than an inherent effect of dupilumab. Some authors have suggested that the blockade of IL-4 and IL-13 may increase the activity of specific ligands (such as OX40 ligand) involved in atopic keratoconjunctivitis. This comment has been added in the discussion section.
Comment: Discussion part: “In our cohort, extrinsic status and baseline total IgE did not correlated to response to dupilumab both at week 4 and at week 16. This is in contrast with the results of Olesen et al. [10] who described high baseline total IgE as predictors of non-response. How to explain this part?
Reply: We deleted the reference to the study by Olesen et al. since we noticed that it could be misunderstood. Indeed, Olesen et al. found that high serum IgE at baseline were predictors for not achieving EASI75 at 1-month follow up but not at 3-month follow-up, suggesting that longer duration of treatment may be required for full effect of dupilumab in patients with more severe atopic dermatitis.
Reviewer 2 Report
The article is well written and of interest to those of us who manage severe atopic dermatitis. It will be enlightening to see future studies that better inform us which patients are more or less likely to respond to dupilumab.
Author Response
We thank the Reviewer for appreciating our manuscript.